

# COVID-19 infection: Disease detection and mobile technology

Jaya Verma[1],* and Amar Shankar Mishra[2],*

[1] Amity University, Noida, Uttar Pradesh, India
[2] CIPL, New Delhi, Delhi, India
* These authors contributed equally to this work.

Corresponding author
Jaya Verma,
jayaverma745@gmail.com

## ABSTRACT

**Background:** A pneumonia outbreak of unknown etiology took place in Wuhan, Hubei province, China and spread quickly worldwide in December 2019. Chinese Center for Disease Control and Prevention identified a novel beta-coronavirus called 2019-nCoV, now officially known as severe acute respiratory syndrome coronavirus 2 (SARS-CoV2) that is responsible for the pandemic. The coronavirus COVID-19 affected 215 countries and territories around the world and more than 99 hundred thousand people at present (*Nature Nanotechnology, 2020*).
At present, there are no specific vaccines or treatments available for COVID-19. However, there are many ongoing clinical trials evaluating potential treatments. At this time the experts recommend precautions such as social distancing, hand washing, and wearing face masks to reduce disease transmission. This review article aims to improve the readers' awareness towards the important role of mobile technology for SARS-CoV-2.
**Methodology:** To achieve this objective, we performed a COVID-19 literature review from various sources that include data from the published articles as well as World Health Organization reports on coronavirus disease and how mobile technology is useful to fight against this disease.
**Results:** Mobile technology can be helpful in mapping disease spread and provides an easy way to provide awareness that promotes safety and adoption of necessary precautions to mitigate and stop community transmission.
**Conclusion:** The spread rate of COVID-19 is very high and until now no vaccines are available to control this disease. To this end we should leverage other avenues such as digital technologies to protect ourselves from this disease. Mobile technology such as smartphones are playing an important role in this pandemic, by launching apps to track coronavirus infected people. These apps are very easy to use and provide self-isolation guidelines as well as other safety tips.

# INTRODUCTION

The coronavirus disease 19 (COVID-19) is a highly transmissible and pathogenic viral infection caused by severe acute respiratory syndrome coronavirus 2 (SARS-CoV-2), which emerged in Wuhan, China and spread around the world (*Tay et al., 2020*).

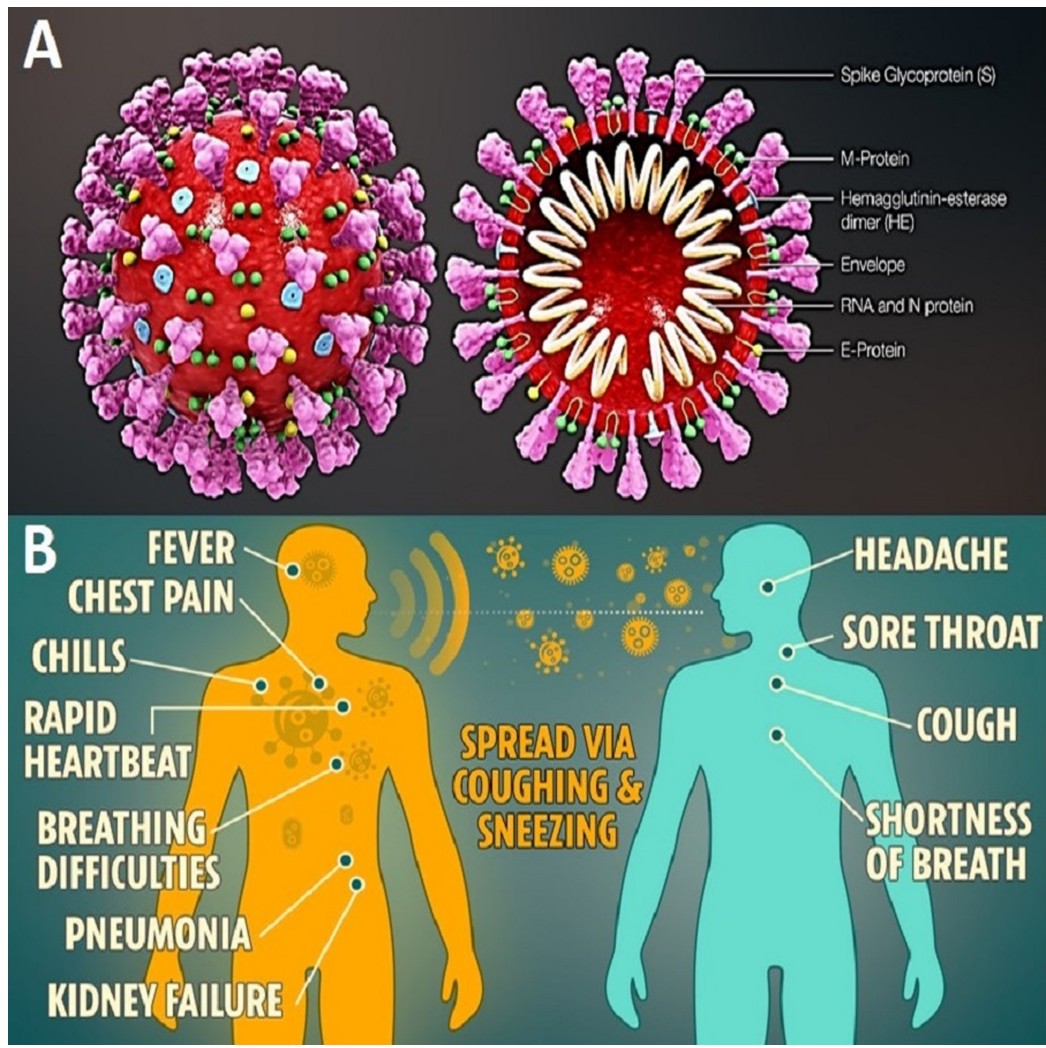

**Figure 1 (A) Structure of corona virus.** (B) The symptoms of COVID-19 & how it spreads.

Coronaviruses belong to the family of Coronaviridae in the Nidovirales order. Corona represents crown-like spikes on the outer surface of the virus (*Wikimedia Commons Contributors, 2020*, https://commons.wikimedia.org/wiki/File:3D_medical_animation_ coronavirus_structure.jpg); thus, it was named as a coronavirus (*Cui, Li & Shi, 2018*). Coronaviruses are minute in size (65–125 nm in diameter) and contain a single-stranded RNA as a nucleic material, size ranging from 26 to 32 kbs in length (Fig. 1A).
The subgroups of the coronaviruses family are alpha (α), beta (β), gamma (γ), and delta (δ) coronavirus. The severe acute respiratory syndrome coronaviruses (SARS-CoV), H5N1 influenza A, H1N1 2009, and Middle East respiratory syndrome coronavirus (MERS-CoV) have been shown to progress in some patients to acute lung injury (ALI) and acute respiratory distress syndrome (ARDS) which leads to pulmonary failure and fatality (*Zhong et al., 2003*; *Wang et al., 2013*; *Shereen et al., 2020*).

A coronavirus is easily transmitted through coughing and sneezing. Due to its high rate of spread, it is important to follow government guidelines to reduce (Fig. 1B) COVID-19 (https://positivebioscience.com). In this case, the best way to protect people; stay home, take precautions, and eat healthily to strengthen our immune system till vaccine development, suggested by WHO report (www.who.int/blueprint/priority-diseases).

## COVID-19 detection

According to the World Health Organization, diagnostic testing for COVID-19 is critical to tracking the virus, understanding the epidemiology, informing case management, and suppressing transmission. *Maxmen (2020)* described the strategic use of diagnostic testing through many in-house and commercial assays to detect the COVID-19 virus. Many of these molecular assay tests are currently being validated. Samples for assay testing can be collected several different sites in the patient. Samples can be taken nasally or via the back of the throat (*Maxmen, 2020*). For patients in the hospital, a sample from the lower respiratory tract may provide the best results. Antigen testing reveals whether someone has a current infection and could therefore pass Covid-19 on to others (*Cookson & Hodgson, 2020*, www.ft.com). In contrast, antibody (or serological) tests uses blood samples to detect the immunity conferred by past infection. The kits used for antibody testing use proteins from the virus as "glue" to trap antibodies present in the blood (*Smriti, 2020*).

## Epidemiological summary

Almost 50 hundred thousand cases of coronavirus disease were reported from 31 December 2019 to 11 May 2020, including around 28 hundred thousand deaths shown in Fig. 2 based on a report conducted by *European Centre for disease control & Prevention (2020)*. Table 1 provides a global representation of the COVID-19 pandemic for the same duration (*Jun et al., 2020*). At present, COVID-19 is a public health emergency of international concern (*Wu & McGoogan, 2020*). While investigations are ongoing, at the time of publication, there is no known specific, effective, proven, pharmacological treatment (*Nature Nanotechnology, 2020*; *Pulla, 2020*). In vitro studies have suggested that chloroquine, an immunomodulant drug traditionally used to treat malaria, is effective in reducing viral replication in other infections, including the SARS-associated coronavirus (CoV) and MERS-CoV (*Meo & Klonoff, 2020*; *Saleh et al., 2020*). However, the efficacy and safety of chloroquine for the treatment of SARS-CoV-2 pneumonia remains unclear. The World Health Organization and other research institutes are continuing on work on vaccine development (*World Health Organization, 2020*). Given the lengthy process of investigating therapies for stopping COVID-19, mitigation strategies remain an important and effective facet in slowing the virus spread. One such mitigation strategy is the use of mobile technology (*Colson, Rolain & Raoult, 2020*; *Savarino et al., 2003*; *Stoye, 2020*).

According to a 2008 Bulletin of the World Health Organization, mobile technology played a significant role during the China earthquake. The 2008 earthquake with a magnitude of 8.0 struck the northwestern region of Sichuan province, China. More than 80,000 people were killed and 5 million more became homeless. An urgent issue associated

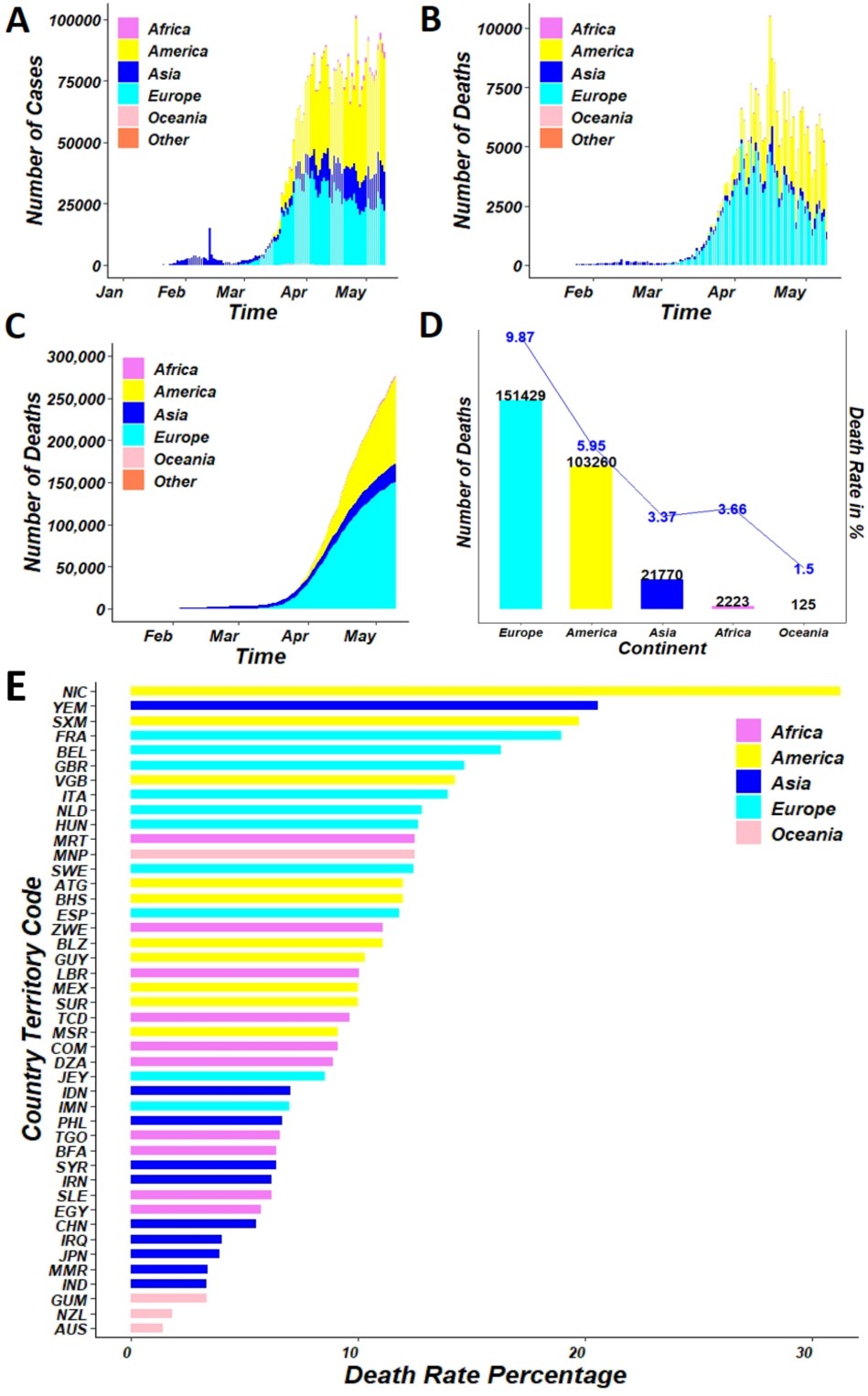

**Figure 2** (A) Day-to-day distribution of confirmed cases (B) distribution of daily death cases (C) distribution of cumulative death (D) comparison between actual death and death rate (E) death rate distribution of top 40 countries; of COVID-19 over the continent, as of May 11, 2020.                             

**Table 1 Global scenario over active and death cases of COVID-19.**

| Rank of continent | Continent | Cases | | | Deaths | | |
|---|---|---|---|---|---|---|---|
| | | Rank | Country | Status | Rank | Country | Status |
| 1 | **America** (Total cases-1,736,710 Total deaths-103,260) | 1 | United States of America | 78,794 | 1 | United States of America | 78,794 |
| | | 2 | Brazil | 10,627 | 2 | Brazil | 10,627 |
| | | 3 | Canada | 4,693 | 3 | Canada | 4,693 |
| | | 4 | Peru | 1,814 | 4 | Mexico | 3,353 |
| | | 5 | Mexico | 3,353 | 5 | Peru | 1,814 |
| 2 | **Europe** (Total cases-1,534,605 Total deaths-151,429) | 1 | Spain | 26,478 | 1 | United Kingdom | 31,587 |
| | | 2 | Italy | 30,395 | 2 | Italy | 30,395 |
| | | 3 | United Kingdom | 31,587 | 3 | Spain | 26,478 |
| | | 4 | Russia | 1,827 | 4 | France | 26,310 |
| | | 5 | Germany | 7,395 | 5 | Belgium | 8,581 |
| 3 | **Asia** (Total cases-645,103 Total deaths-21,770) | 1 | Turkey | 3,739 | 1 | Iran | 6,589 |
| | | 2 | Iran | 6,589 | 2 | China | 4,637 |
| | | 3 | China | 4,637 | 3 | Turkey | 3,739 |
| | | 4 | India | 2,109 | 4 | India | 2,109 |
| | | 5 | Saudi Arabia | 239 | 5 | Indonesia | 959 |
| 4 | **Africa** (Total cases-60,661 Total deaths-2,223) | 1 | South Africa | 186 | 1 | Egypt | 514 |
| | | 2 | Egypt | 514 | 2 | Algeria | 494 |
| | | 3 | Morocco | 186 | 3 | Morocco | 186 |
| | | 4 | Algeria | 494 | 4 | South Africa | 186 |
| | | 5 | Ghana | 22 | 5 | Nigeria | 128 |
| 5 | **Oceania** (Total cases-8344 Total deaths-125) | 1 | Australia | 97 | 1 | Australia | 97 |
| | | 2 | New Zealand | 21 | 2 | New Zealand | 21 |
| | | 3 | Guam | 5 | 3 | Guam | 5 |
| | | 4 | Northern Mariana Islands | 2 | 4 | Northern Mariana Islands | 2 |
| 6 | **Others** (Total cases-696 Total deaths-7) | 1 | Cases on an international conveyance Japan | 696 | 1 | Cases on an international conveyance Japan | 7 |

with the aftermath of the earthquake was the efficient detection of occurrences of epidemic-prone diseases so that quick action could be taken to prevent outbreaks. Before the earthquake, the local health-care agencies were required to report 38 types of infectious diseases, as mandated by the law on prevention and treatment of infectious diseases, through the Chinese information system for disease control and prevention (CISDCP) to a national database (*Ma, Yang & Shi, 2006*). In Sichuan, an electronic dial up/landline internet-based disease surveillance system has been in place in all townships since 2004. However, the earthquake paralyzed much of the traditional landline-based infrastructure in many affected areas. While working to repair the landline-based reporting system, the China CDC (Chinese Center for Disease Control and Prevention) developed an emergency reporting system based on mobile phones. *Yang et al. (2009)* explained the system and the lessons learned from the utilization of mobile phones for infectious disease
surveillance after the catastrophic earthquake. The current surveillance system leveraging mobile phone technology has played an important role during COVID-19 (*Stoye, 2020*).

In this pandemic, digital technologies like smartphone applications (apps) using Bluetooth technology are needed to track infected people in nearby areas (*UNESCO, 2020*, en.unesco.org). Such apps are developed worldwide by the USA, Singapore, India, UK, and many other countries to track and control the coronavirus disease (*Guermazi, 2020*). These apps provide self-quarantine and other safety information to users. The authors argue that if greater numbers of people downloaded and used these sorts of apps, there is potential for decreasing the spread rate of the coronavirus disease. A detailed study of these smartphone apps are discussed in this article within the context of the COVID-19 response.

## LITERATURE REVIEW METHODOLOGY

This review article has been structured after collecting data from COVID-19 published articles from sources including but not limited to from nature.com, Elsevier, science direct.com, Royal Society of Chemistry publishing articles, and American Chemical Society articles. *Shereen et al. (2020)* discussed COVID-19 infection, its origin, transmission, and characteristics. Some of major supportive literatures for this article are; *Colson, Rolain & Raoult (2020)* addressed treatment and vaccine progress for coronavirus. *Morley et al. (2020)* described ethical guidelines for contact tracing apps. *McKendry (2020)* presented evidence that "apps for COVID-19 contact-tracing are secure and effective". These articles have played an important role in the literature review for this article. In addition, the authors used Google to identify the top 10 mobile apps used track COVID-19 which revealed *Chaturvedi (2020)*, *Gould (2020)* and *Wakefield (2020)*.

## SMARTPHONE TECHNOLOGY TO FIGHT AGAINST COVID-19

Smartphone apps are playing an important role in the response to the Covid-19 pandemic (*Yang et al., 2009*). These apps are being used to track infected people, issue self-quarantine guidelines, provide the latest communication to the citizens and ease the burden on healthcare staff throughout the world (*Ma, Yang & Shi, 2006*). The apps have been downloaded by millions of people. Technology is providing an important role in the diagnosis of those affected, identifying hotspots, and providing real-time information updates (*Villa, Sankar & Shiboski, 2020*). This article, provides discussion of popular smartphone apps specific to tracking the Covid-19 outbreak (www.geospatialworld.net/popular-apps-covid-19).

### TraceTogether

TraceTogether is a popular smartphone contact tracking app that uses Bluetooth to track infected people and alert people who have been close to them in the past 15 days. Anyone with a Singapore mobile number and a Bluetooth enabled smartphone can download this app (*Koh, 2020*).

The application was developed by the Government Technology Agency (GovTech) in collaboration with the Ministry of Health (MOH) and has become a prototype for many

other contact tracking applications in other parts of the world. When two people using the app are close, both phones will use Bluetooth to exchange a temporary ID. This temporary identification is generated by encrypting the identification of the user with a private key held by the MOH. The MOH can only decipher it and does not reveal its identity or the identity of the other person. This application does not collect data on the position of the GPS or the Wi-Fi/mobile network.

## Aarogya Setu

In this app, monitoring is done via Bluetooth and a location-generated graph that records proximity to any infected person. This app has been developed by the Indian Ministry of Electronics and it notifies users if they have crossed paths with someone who has been diagnosed positive for Covid-19 (*Business Today, 2020*). India is also using various other apps such as "Kerala solutions", "Tracking quarantine", "More than just tracking" to track and trace Covid-19. These apps are cross-communicate to each other and would be helpful for entire country people, who are using these apps.

## COVID symptom tracker

Scientists analyzed the high-risk areas in the United Kingdom, the rate of spread of the virus, and the most vulnerable groups, depending on health conditions to develop this app. "Covid Symptom Tracker" was designed by doctors and researchers from King's College London and St. Thomas hospitals, in association with a private health company called Zoe Global (*Wakefield, 2020*). This app monitors virus symptoms for ongoing research and also tracks virus among those using the app. The app complies with the general data protection regulation and the data is used only for health research and not for commercial purposes.

## The corona dataSpende

This is a German smart-watch app and monitors the spread of coronavirus by collecting vital signs (heart rate, sleep patterns, body temperature) from volunteers using a smart-watch or physical activity tracker (*Busvine, 2020*). This app can check whether a person has developed symptoms of Covid-19 or not. The results are displayed on an interactive online map that allows health authorities to take stock of the situation and determine hotspots.

## CovidWatch

This application allows people to protect themselves and their communities while ensuring privacy. This app is designed in collaboration with Stanford University (*COVID Watch, 2020*). It uses Bluetooth signals to detect users when they are nearby and warns them anonymously if they were in contact with someone who had tested positive. It was one of the first applications to launch an open-source protocol for decentralized tracking of Bluetooth contacts that preserves privacy. A distinctive feature of the app is that any third party, including the government, will not be able to track who has been exposed by whom.

### NHS smartphone app

This contact tracking application, developed by the British National Health Service. The app was designed by NHSX (the NHS innovation unit) and will be released in the near future (*Gould, 2020*). The app will maintain control of people's movements and alert those who come into contact with those who have been infected. The NHS suggests that by analyzing virus spread patterns and hotspots, the app would also help in relaxing lockdown. The app would classify user details based on demographics, home structures and mobility patterns. Based on the data analysis a maximum number of people could be established and allowed to move freely. British health secretary Matt Hannock urged the public to download the app as soon as it becomes available.

### Let's beat Covid-19

This app was developed by MedShr, used by more than one million physicians in diagnosis of Covid-19 (*Butcher, 2020*). LetsBeatCOVID.net is designed to allow members of the public to complete a short survey on their health and exposure to COVID-19 so that health services can identify at risk individuals. The public is invited to complete a brief anonymous survey of them and is also allowed to enter information about their family members.

### HaMagen

This application was launched by the Israeli Ministry of Health and uses contact monitoring to contain the spread of deadly infection (*Winer, 2020*). The application lets users to know if they have been close to someone diagnosed with the virus in the past 15 days. Once a user installs the application their movements are tracked using location technology, and the information obtained is compared with the ministry's data on that particular location for diagnosed people. If a particular user was very close to an infected person, the app redirects the person to the Ministry of Health website where they can register for quarantine.

### Kwarantana Dommowa

Poland was one of the first western countries to launch a smartphone app that collects a great deal of personal information, including the location of people and digital photos (*APK Center, 2020*). When using this app, people upload their selfie image when requested by app administrative agents, so they can determine their exact location. This app is mandatory for anyone who has developed coronavirus symptoms in Poland.

### PeduliLindungi

This application has been developed by the Indonesian Ministry of Communications and information, together with the Ministry of State Societies (SOE). This application allows users to collect data related to the spread of COVID-19 in their communities and help strengthen government efforts to track confirmed cases as well as those suspected of being infected with the virus (*Novianty, 2020*). When a user is close to another user whose data has

been uploaded to PeduliLindungi, the app allows anonymous identity exchange, according to its official website.

## Limitations of smartphone technology

A major limitation is that any contact tracing plan must reach a critical mass in order to be effective. People need to both download an app and update their real health status through any given downloaded app (*Eddy, 2020*). Success in terms of health status data collections depends on messaging and how it is presented to the user. Messaging should emphasize that apps provide disease mitigation and protection for individual users and others at large (*Joshi, 2020*).

Mobile app technology related to Covid-19 or other epidemic and pandemic diseases requires data aggregation from multiple smartphones to compute intersections of trajectories. Such aggregation will be hard to implement at scale if such app use remains decentralized and centralization will require additional infrastructure. Even with centralized aggregation, rigorously estimating the dynamic network parameters and the associated error models will be a non-trivial task, especially without near-universal participation (*Hausfeld, 2020*). At best, mobility data may be used for modeling macro-level patterns of infection spread that too with several simplifying assumptions with uncertain error models (*United Nations, 2020*). Besides, making such apps universal, and centralized aggregation with support from mobile service providers, Google, and indoor Wi-Fi providers, will certainly be beyond individual app developers and will require governmental support (*Zargar, 2020*). China leveraged facial recognition technology, with existing infrastructure that was already in place. While this tactic has been successful in China, there are serious privacy and data protection concerns that need to be addressed— in terms of legitimacy and proportionality, regulatory oversight, access control, and purpose limitation (*Morley et al., 2020*).

Each presented and future app-based technology specific to Covid-19 and other epidemic and pandemic diseases presents with both advantages and limitations. These advantages and limitations must be rigorously evaluated and taken into account when choosing the best app to meet disease process and population needs (*Grout, Weaver & Doyle, 2020*).

## DISCUSSION

Researchers around the world are rushing to create vaccines and medicines that can stop the COVID-19 pandemic or at least halt its spread. In the midst of these efforts, there is evidence that technology can play a useful role in mitigating the crisis and facilitate a valuable contribution to this global battle (*Ting et al., 2020*). The use of mobile devices as part of this effort has raised several important questions around privacy and security (*Xiao & Fan, 2020*, *Nature, 2020*). COVID-19 tracking apps fall into three main categories: (1) understanding general population movement, (2) potential proximity to COVID-19 positive individuals and advice on measures for self-quarantine, and (3) the collection of information from patients for statistical analysis (*UCLG, 2020*).

## Mobile tracking to understand population movement and the impact of lockdown

Mobile carriers in Germany, Italy, and France have started to share mobile location data with health officials in the form of aggregated, anonymized information and is consistent with local law and regulations. Because European Union member countries have very specific rules about how any app and device users must consent to the use of personal data, developers must consider other forms of useful data unless they solicit and receive individual consent from users. The aggregated and anonymized approach is related to groups within a population and not individuals, but it can provide a clear view of population displacement trends and therefore disease transmission risk level of geographic areas (*Grout, Weaver & Doyle, 2020*).

## Determining potential proximity to COVID-19 positive individuals

This approach is being explored in countries such as Germany and France. The objective is to limit the spread of the virus by identifying people who have potentially come into contact with an individual who has tested positive, and by advising those people to self-quarantine, if proximity was determined. In Germany, the government is relying on the rules defined by the Pan-European Privacy-Preserving Proximity Tracing. France is exploring this subject with INRIA (The National Institute for Research in Computer Science and Automation) under the project: ROBERT (ROBust and privacy-preserving proximity tracing) 260 protocol (*McKendry, 2020*).

## Collection of users' information for statistical analysis

This approach has been used by the UK government through the application "C-19 COVID Symptom Tracker", which was developed by the startup ZOE in association with King's College London. The data needed to meet all three objectives are then stored by mobile providers in a variety of places that must be secured, both to protect the app users' privacy but also to prevent manipulation/spoiling of the data by a third party. In this case data is sourced from different places, like repositories of GPS, Bluetooth, and other apps on the device, different security arrangements by the source may need to be considered (*Morley et al., 2020*).

Regulators are recognizing that app developers need timely guidance to balance the collection of data with safeguarding privacy. In the EU, the statement by the EDPB Chair on the processing of personal data in the context of the COVID-19 outbreak, published in March 2020, advances this objective (*UCLG, 2020*). In this field of research, app providers must to ensure an appropriate level of security, to avoid any data leaks and any data manipulation by non-trusted third parties. App developers should build in the ability to discontinue their use if national health authorities determine that the data they collect is no longer needed to address the pandemic (*Rocher, Hendrickx & de Montjoye, 2019*).

## CONCLUSION

The coronavirus is believed to have started to spread from the Hunan seafood market at Wuhan, China and quickly spread up to 215 countries. While various clinical trails

have begun as it relates to vaccine availability and treatment therapies, at present, there remain no approved evidence-based vaccines or therapies for the treatment human coronaviruses, specifically Covid 19. Scientists and researchers are continuously working to develop efficient therapeutic strategies to cope with the COVID-19. There are numerous organizations working towards the advancement of successful SARS-CoV-2 vaccines, but these vaccines still require months for commercialization after rapid human and animal-based successful trails. In the meantime, control of virus spread remains paramount. Mobile technology, app-based technology is playing an important in various countries by tracking virus spread and providing information related to best-practices in mitigations such as self-quarantine. These apps are easy use and successful broad adoption will likely increase as the literature begins and continues to report the effectiveness of this technology (*Joshi, 2020*). As the saying goes, "a crisis provides an opportunity"; this first great crisis of 2020 provides an opportunity to establish best practices in the use of mobile technology for healthcare purposes. The potential benefits of digital app-based healthcare interventions seem particularly compelling for managing chronic conditions such as diabetes and hypertension (*Huckman & Stern, 2018*).

## ACKNOWLEDGEMENTS

We would like to thank Eric B. Bauman, Lisa Buckley and Arun Mathews for their comments, insightful suggestions, and careful reading of the manuscript.

### Funding

There was no additional external funding received for this study.

### Competing Interests

Amar Shankar Mishra is employed by CIPL, New Delhi and the authors declare that they have no competing interests.

### Author Contributions

- Jaya Verma conceived and designed the experiments, performed the experiments, prepared figures and/or tables, authored or reviewed drafts of the paper, and approved the final draft.
- Amar Shankar Mishra analyzed the data, prepared figures and/or tables, authored or reviewed drafts of the paper, and approved the final draft.

### Data Availability

Covid-19 raw data as per the European Centre for Disease Prevention and Control (ECDC) used in Fig. 2 is available as a Supplemental File.

## Supplemental Information

Supplemental information for this article can be found online at http://dx.doi.org/10.7717/peerj.10345#supplemental-information.

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
