# Peer review of "COVID-19 infection: Disease detection and mobile technology"

_PeerJ, doi:10.7717/peerj.10345_

## Round 0.1 · original submission · Major Revisions

Your manuscript is novel and timely. Reviewers find the topic and your work to be translational and with significant potential. Please address the comments and suggestions made by both reviewers. Please pay specific attention to decreasing the background of the psychophysiology and etiology of COVID-19 and place more emphasis on the novel aspect of your work and this manuscript. Focus on how the technology, apps were used, why this is novel and provides translational solutions.

·

Basic reporting

1. Basic Reporting
Clear English-
• Needs to be improved. Line 16 need an additional word between world and more. “…territories around the world affecting more than 48 hundred…”
• Take out additional words that have no meaning such as Line 16 “At this time, there are no specific vaccines or…”
• Line 17-18 needs re-wording. Example, line 18, “At this time the experts recommend precautions such as social distancing, hand washing and wearing face masks to reduce disease transmission.”
• Line 19 needs re-wording. “…is to improve the readers awareness of the importance of…”
• Strange capitalization of letters, wuch as line 23, “It” should not be capitalized as well as “Mobile”.
• Line 24 reword…”it can help to aware” is not correct English.
• Line 25 take out “to an extent.”
• Line 33 should say “transmissible” instead of transmittable
• Line 35 re-word for better English.
• Line 41, change cause to “have been shown to progress in some patients to…”
• Line 42 take out “results in”
• Line 44 needs re-working
• Line 46 take out “among peoples”
• Line 51 re-word
• Line 52- does it need to be via Bluetooth or just via mobile device?
• Line 53 poor English
• 54 use the term applications instead of “apps” which is a shortened form of the word. You can use (apps) as an abbreviation throughout the manuscript if you introduce it in that way.
• Line 56, start by writing out these locations with the abbreviation in parenthesis, then later in the paper you can use the abbreviations
• Line 68, Diagnostic should not be capitalized
• Line 125, once you specify (MOH) you can then use that abbreviation in line 126.
• Line 130, “Monitoring” should not be capitazlized
• Line 139, what is GDPR?
• Line 142 should say “vital signs”
• Line 144 needs re-wording for better English

Clear writing- I would recommend hiring someone who can assist with your English. I understand that it is a difficult language to write in, however, it is essential to have good English in order to get your message across.

Literature references
• Need more references, such as line 50 where it says it was suggested by WHO but there is no referenced listed.
• Several more sentences through the description of the applications need to be cited.
Sufficient background/context provided
• Too much background on the testing. Just a short background on the virus itself can introduce the topic
• I would add how mobile apps have helped in other disease outbreaks in the past in your background section
Professional article structure, figures and tables
Raw Data shared? None
Is it broad and cross-disciplinary? It is broad.

Experimental design

2. Study Design
Article content aligns with the scope/aims?
• Line 20, I would be more specific on the literature review you performed. It seems that it would be around mobile technology and COVID-19 and not just COVID-19?

Rigorous investigation performed to high technical and ethical standards?
• This is hard to evaluate based on the methodology described in the paper. It appears that you did a good search and evaluation of the applications but I just need how you did it to be more specific.
Methods are in alignment, described in detail and able to be reproduced? No
Methodology consistent with the content
• Your methodology should either be a systematic review or a different kind of review. I would clarify this in line 20-21 where it says “a form of systematic review”, but it doesn’t specify any form other than a general literature review which is a very broach term for reading literature. Additionally, I would add in the methodology section here how you searched the literature including databases used and search terms. Lines 59-66 I have the same comment. This is not a reproducible search- need to be clearer on what search terms you used in which databases. The Google search seems clear to me.
• Line 23 “This Mobile technology” does not tell us what mobile technology you are referring to and needs to be re-written to be clear.
• Line 68-92 seem unrelated to your research question about which mobile apps are available and useful
Sources adequately cited?
• No, line 111 needs a reference
• No, line 112-113 needs a reference
• No, line 114 needs a reference
• No, line 115 needs a reference
• No, line 119 needs a reference
• No, line 205-207 needs several references
• No, line 208 needs a reference
• You don’t need to acknowledge the WHO for their information, just properly cite them in your paper
Is it organized logically?
• No, lines 94-110 belong in a background/intro statement.
• 162-163 should be moved up to the beginning of this App description.

Validity of the findings

3. Validity of Findings
Impact and novelty assessed? Yes, this is a novel and good idea for a paper.
Conclusions well stated?
• The conclusion starts with a statement that is novel to the manuscript on a topic that is unrelated to the apps.
Speculation is identified as such?
• Line 195-196 could be seen as speculation. Is there evidence that these apps protect people? If so, cite it here. Otherwise you’ll need to explain your position.
Well designed argument that aligns with the Intro?
The conclusion identifies unresolved questions/gaps/future directions? No, future research directions should be identified and limitations of your paper should be identified (not limitations of the apps).

Additional comments

4. General Comments
I would completely take out the section on testing and the related Figures. This does not relate to your paper which is aiming to focus on reviewing the available COVID 19 apps and their uses. I would expand of the use and function of the applications you described. I would try to organize the apps by location or at least state that it is a global perspective and that you will be reviewing one app from 10 (or however many use have) different countries to give an overview of the landscape of the current available apps.

·

Basic reporting

a) I would consider this a traditional or narrative literature review

My expectation is that it:
1. Critiques and summarizes a body of literature
2. Draws conclusions about the topic
3. Identifies gaps or inconsistencies in a body of knowledge
4. Requires a sufficiently focused research question

Broadly speaking, I believe that 1 and 2 have been accomplished here, however, 3 and 4 were somewhat vague.

b) The data collection detailed on lines 60 and 61 referencing articles Sheeran and Colson et al needs to be expounded upon further.

c) I would be grateful for more explanation around the rigor behind selecting the COVID mobile technologies. What were the sub-types of app? (i.e., geolocation based contact tracing, Currently, it feels like a fairly simple search engine query was primarily utilized. this is not problematic in an of itself, but if so, I'd like to know the rationale for the search term string(s).

d) Sections 2.1-2.3 seem unnecessary and in my opinion, take away from the main premise of the paper. Perhaps condensing it into a simpler summary?

Experimental design

a) The survey methodology, as stated above, seems limited and I would appreciate greater clarity around how these particular apps were selected.

b) This should be further catagorized into by the types of technology used by the app and the strengths and weaknesses of each.

c) The most significant area appears to also be the most underdeveloped: The limitations of each type of technology. This needs to be developed robustly or at least discussion should occur as to what areas need further research? Examples might include the limitations of bluetooth based contact tracing, as well as practical implications for battery life.

d) Of particular interest is the privacy implications of gathering this location and contact tracing data. This may be challenging as privacy laws differ based on state / country, however, a general discussion as to the challenges and concerns citizens may want to consider with the use of these apps might be helpful.

Validity of the findings

The information presented here appears helpful and may be valid to a certain degree. Given the lack of explanation around the search methodology, it would be difficult to come to the conclusion that this is a comprehensive list of apps.

Furthermore, the efficacy of the apps in tying into public health initiatives and subsequent reduction of spread is what would send this article into the stratosphere in terms of usefulness.

Additional comments

An earnest first draft.

I would reduce the amount of unnecessary literature review around testing, and expand the discussion around the methodology behind why these apps where chosen to be studied.

Consider expanding the section around limitations and tie this to the inherent limitations of the various forms of technology.

Consider really understanding the privacy concerns (personal health information and location tracking) associated with these applications and sharing these with the reader.

Consider explaining what the limitations of this review are and what next steps might look like for further research.

---

## Round 0.2 · Major Revisions

You manuscript has undergone an additional review as a result of your recent and considerable revision. Reviewers continue to find your manuscript timely and of interest in ways that are translational and contribute to the literature. I do believe the resubmitted manuscript is much improved. I would urge you to closely follow the recommendations of the most recent reviewer (reviewer #3).

·

Basic reporting

1. The topic is relevant and timely. What is needed throughout the manuscript is clear dates for data. With the rapidly changing information, what is written within the manuscript has already had changes. If this article is to be used to guide decisions for future work or citations, it will be imperative that the dates of information retrieval are given.
2. The epidemiological summary moves from a broad overview of the disease, the current efforts to create a cure, and then move into technology; this is great. However, the move to using technology requires a smoother transition. Currently lines 82-83 say the easiest way to protect ourselves is using digital technologies. This is written as if the app itself has a healing property, which is not the case. Perhaps transitioning to the discussion of technology can be done by saying something like “Given the lengthy process of investigating therapies for stopping COVID-19, mitigation strategies are proving to be effective in slowing the spread. One such strategy is the use of mobile technology.” Then move into the discussion starting on line 85 about previous uses of technology in disaster response.
3. The limitation of smartphone technology (section 3.11) does not flow well. Lines 223-224 and 234-235 are confusing to the reader. When reading I see there are limitations in a few areas: the fact that success of the app in mitigating the spread is dependent on large numbers of downloads and user input, and data aggregation. Perhaps separate these limitations into paragraphs for easier reading. The discussion on impact of the app on battery life does not add value to this section. One area that could be further explored is the collection of personal information, it’s mentioned briefly but not really unpacked.
4. The entire manuscript requires edits for English language flow. To point to some specifics: lines 24-31, lines 46-50, line 196, line 201.
5. Line 207 should read: “Information, together with the Ministry of State Societies (SOE). The application allows user to”
6. Line 210 does not seem to connect to anything, it’s confusing.
7. It is not clear who the article is intended for, is it for healthcare personnel? Epedimiologists? Educators? Public Health officials?

Experimental design

1. The aim is clear, but could be more focused. It’s currently stated that the aim is to improve the readers awareness of the important role of mobile technology for SARS-CoV-2. After reading the manuscript, I believe this can be narrowed to specifically focus on mobile applications role for SARS-CoV-2 (rather than mobile technology in a broader sense).
2. The methodology does not provide a clear description of what was done. It is hard to say if this was a comprehensive review. What were the search terms, the date ranges, the number of articles found? As this currently reads, there is not enough detail to replicate this review as the use of technologies and the pandemic progress.
3. The apps that you chose to highlight, were these all that were discussed in the literature? Where there some apps that you decided not to discuss (if so, this should be stated and why)? Did the literature show any data in the effectiveness of tracking and preventing? If so, these are things that should be discussed in the manuscript.
4. Lines 180-181, the sentence “The contact tracking application, currently developed by the National Health Service, which is the national health system funded by England” is a repeat from earlier in the paragraph and should be removed.

Validity of the findings

1. I do not see a clear conclusion that wraps up the manuscript and links it to the aim of the literature review.
2. Lines 290-292 stating “The easy use and successful application of mobile technology to fight against COVID-19 pandemic will probably increase the extreme public acceptance soon.” This is speculation and should be noted as such, there is no data in the manuscript to support this claim.
3. Jumping from the use of an app for COVID monitoring chronic diseases in the future is disjointed and leaves the reader hanging at the end of the article.

---

## Round 0.3 · Minor Revisions

Please pay careful attention to the several recommendations that Reviewer Three has provided in the current iteration of your manuscript, including their direction as follows. "My previous comment of jumping from the use of an app for COVID to monitoring chronic diseases in the future is disjointed still stands".

In addition please carefully review the manuscript for conventions related to English language publication. Thank you for your continued patience and the continued improvements of your manuscript. The topic is timely and innovative.

·

Basic reporting

1. Throughout the manuscript "this study" or "this article" is used. It's unclear if the meaning is studies/articles reviewed from published literature or the manuscript we are reading. Examples: line 54, "this study describes the strategic use..." there is no citation and it's unclear if "this study" is the current manuscript. Also occurs in line 97 "this paper", line 105 "this article". Line 109 "this review article" is assumed to mean the manuscript we are currently reading, but the confusion in using "this" earlier in the paper leads me to be unsure.

2. Line 147-149 are not understood by the reader. the sentence seems to start out talking about particular apps, but then ends with "No involvement of epidemiologists a problem", is this an app name? Or are you stating that there is no problem when epidemiologists are not involved with the data?

3. There should be a transition before section 3.3 to let the reader know you are moving on to talk about specific apps you found in the literature.

Experimental design

No comment

Validity of the findings

1. I see you added in references for the claim stating "The easy use and successful application of mobile technology to fight against COVID-19 pandemic will probably increase the extreme public acceptance soon.” You added the reference in the limitations of smartphone technology (section 3.11). I would reference the article where you make the claim.

2. My previous comment of jumping from the use of an app for COVID to monitoring chronic diseases in the future is disjointed still stands.

---

## Round 0.4 · Minor Revisions

The overall content and structure of this manuscript is much improved from previous iterations. However the manuscript will still requires additional careful attention to copy editing.

---

## Round 0.5 · Minor Revisions

Dr. Verma - I continue to believe that this manuscript is interesting and timely. In general, I leave the much of the editing suggestions to the reviewers, but in this case, the topic of mobile technology is of significant interest based on my portfolio of research and publications. To this end, I have uploaded the manuscript with suggestions in "track changes". You are of course free to accept them or not, but I do urge you to consider them.

In addition there are several comments where I have suggested you provide a citation or where I think the writing needs further clarification. In terms of clarification, please see your discussion on centralization and decentralization. I would try to make this more succinct and relatively brief - a sentence or two given it is not something you have highlighted elsewhere in the manuscript.

Thank you for your continued patience as it relates to this manuscript.

---

## Round 0.6 · Minor Revisions

Please pay special attention to the recommendations from reviewer Number1. Your manuscript may benefit from additional professional copy editing.

·

Basic reporting

Basic Reporting
• Clear, profession English
o Line 13 should say “that is responsible”
o Line 16 should say “treatments are available”
o Line 17 should say “trials are evaluating”
o Line 22 should say “data from the published articles”
o Line 25 should say “precautions to mitigate”
o Line 41 should say “coronaviruses.”
o Lines 46-48 should be re-written, perhaps something like “Coronavirus is easily transmitted through coughing and sneezing. Due to its high rate of spread it is important to follow government guidelines to reduce transmission.”
o Line 57-58 should be reworded to say “Samples can be taken nasally or via the back of the throat.”
o Line 76 should say “institutes are continue”
o Line 103 should say “these sorts’ apps” and “spread rate of”
o Line 104 should say “within the context”
o Line 106, I’m thinking this should say “Literature Review Methodology” instead of “Survey Methodology”
o Line 114-115 should end with “10 mobile apps used to track COVID-19. The rest of line 115-116 should be moved to the findings of the literature review section.
o Line 146 there is no need for a ‘ before Covid.
o Line 172-173 should say “Based on the data analysis, a maximum…”
o Line 96-97 need a citation.
o Lines 109-112 need more explanation and integration of the literature review to the current research question. How did they guide the review or inform the results of this review? The discussion of what was found in the articles of the literature review should be moved to a later section instead of in the methodology.
o End of line 121 needs a citation.
o Lines 127-203 Overall, each line that is talking about an app needs to have a citation.
o Line 209 should say “provide disease mitigation” and “and the population at large”
o Line 213 should say “implement at scale if such app use remains decentralized and centralization…”
o I would take out the sentence in line 234-235 that start with First. I would start the next sentence with “COVID-19 tracking apps fall into three main categories…”
o Line 267 needs should say “providers must to ensure”
o Lines 142-143 I’m not clear on the point of you mentioning these other apps, I would reword this to be more clear. Perhaps you want to say that it is better for an entire country to use one app or comment if the various apps cross-communicate with each other. I think this needs more explanation.
o Line 257-259 needs citation.
o I’m not sure which citation style you are using, so it is difficult for me to comment on if the citations are correct or not. My personal limitation is that I’m only really familiar with APA and Vancouver, neither of which are being used in the paper. Use of a free citation manager such as Mendeley or EndNote would assist if you are not already using one.
o Line 272 I would not introduce new information here since it is the conclusion, so perhaps just restate the extent of the covid-19 spread and impact on the word here instead of the first sentence.
o Line 277, take out “Our” and just start the sentence with “Scientists”
o To me, figure 2 is not necessary because it is not adding to your research question of how mobile apps are useful. I do understand how Figure 1 and 3 are giving background information about the virus.

Experimental design

I do not have any issues with your study design. The methodlogy is clear and I think the overview of apps is an important topic.

Validity of the findings

Just see my comments about citations about. I think your conclusions/discussion make sense in your paper, just be sure you are citing every sentence that has come from the literature.

Additional comments

Thank you for sharing your work and graciously accepting my edits. Overall, I think the paper reads better and is more focused than the first time I reviewed it. I think the topic is important. While I realize how difficult it can be to try to write in English when it is not your first language, it is extremely important that it reads and flows well with professional English, hence my comments and edits provided above. I would recommend hiring a native English speaker to review you paper after you make further edits. This will give you even more of a polished finish and assist you getting this paper published. I hope my comments are helpful.

·

Basic reporting

Please be sure to double check your within text citations. There are a few that have the author's first name and not the last name (ie. Angela in the abstract).

This edited version reads much easier than previous, thank you for the work put into the readability of the manuscript.

Experimental design

No comment.

Validity of the findings

I appreciate the inclusion of more app examples and their uses. The technology's use in the prevention of spread and the collection of data is much more clear in the revisions.

The conclusion wraps it up well. As I've stated in previous reviews, I would suggest not including the last sentence of the conclusion which introduces the use of apps for chronic diseases, this is not what the article is about.

---

## Round 0.7 · Minor Revisions

Please see the very minor edit requests from the current reviewer so we may proceed to approval for publication.

·

Basic reporting

• Line 75 “continue” should be “continuing on work on vaccine development”.
• Line 107/108 I think RSC and ACS should be spelled out (unless they were already spelled out earlier in the article, but I didn’t see that).
• Line 113 should say “important role in the literature review…”
• Line 114 I’m not entirely clear if this is your intention, but I think it should say “track COVID-19 which revealed www.geospatial...”
• Line 121 should say “in the diagnosis of those affected”
• Line 253 Assuming that INRIA is an acronym, it should be spelled out , same for ROBERT-ROBust in line 254

Experimental design

No comments.

Validity of the findings

No comments.

Additional comments

Overall, your manuscript has come a long way from what you initially submitted. Now, it reads well and make a succinct point about the utility of mobile apps in relation to COVID19. Thank you for the opportunity to review your paper and thank you for graciously accepting my edits and comments.

---

## Round 0.8 · accepted · Accept

Again thank you for your patience and perseverance with the review process. It was a pleasure working with you to see this manuscript progress towards publication.